

# Accelerating compound flood risk assessments through active learning: A case study of Charleston County (USA)

Lucas Terlinden-Ruhl[1,2], Anaïs Couasnon[2,3], Dirk Eilander[2,3], Gijs G. Hendrickx[1], Patricia Mares Nasarre[1], and José A.Á. Antolínez[1]

[1]Delft University of Technology, Delft, The Netherlands
[2]Deltares, Delft, The Netherlands
[3]Vrije Universiteit Amsterdam, Amsterdam, The Netherlands

**Correspondence:** Lucas Terlinden-Ruhl (lucas.terlindenruhl@gmail.com)

**Abstract.** Flooding is the most likely natural hazard that affects individuals and can be driven by rainfall, river discharge, storm surge, tides, and waves. Compound floods result from their co-occurrence and can generate a larger flood hazard when compared to the sum of the individual drivers. Current state-of-the-art stochastic compound flood risk assessments are based on statistical, hydrodynamic, and impact simulations. However, the stochastic nature of some key variables in the flooding process is often not accounted for as adding stochastic variables exponentially increases the computational costs (i.e., the curse of dimensionality). These simplifications (e.g., a constant flood driver duration or a constant time lag between flood drivers) may lead to a mis-quantification of the flood risk. This study develops a conceptual framework that allows for a better representation of compound flood risk while limiting the increase in the overall computational time. After generating synthetic events from a statistical model fitted to the selected flood drivers, the proposed framework applies a Treed Gaussian Process (TGP). A TGP uses active learning to explore the uncertainty associated with the response of damages to synthetic events. Thereby, it informs on the best choice of hydrodynamic and impact simulations to run to reduce uncertainty in the damages. Once the TGP predicts the damage of all synthetic events within a tolerated uncertainty range, the flood risk is calculated. As a proof of concept, the proposed framework was applied to the case study of Charleston County (South Carolina, USA) and compared with a state-of-the-art stochastic compound flood risk model, which used equidistant sampling with linear scatter interpolation. The proposed framework decreased the overall computational time by a factor of four, and decreased the root mean square error in damages by a factor of eight. With a reduction in computational time and errors, additional stochastic variables such as the drivers' duration and time lag were included in the compound flood risk assessment. Not accounting for these resulted in an underestimation of 11.6% (25.47 million USD) in the expected annual damage. Thus, by accelerating compound flood risk assessments with active learning, the framework presented here allows for more comprehensive assessments as it loosens constraints imposed by the curse of dimensionality.

## 1 Introduction

Flooding has been identified as the most likely natural hazard to affect individuals (UNDRR, 2020). Moreover, climate change is expected to increase the magnitude and frequency of extreme water levels (e.g., Hirabayashi et al., 2013; Blöschl, 2022),





which are driven by precipitation, river discharge, surge, tide, and waves (e.g., Couasnon et al., 2020; Hendry et al., 2019;
Parker et al., 2023; Ward et al., 2018). Low-lying coastal areas are especially susceptible to compound events of these drivers,
which enhance the flood hazard (Wahl et al., 2015). In addition, migration patterns are causing an increase in assets and people
to these areas (e.g., Swain et al., 2020; Neumann et al., 2015). Governing bodies tackle the challenge of compound flooding
using the concept of stochastic flood risk (e.g., Klijn et al., 2015; Muis et al., 2015). This concept takes into account the three
following metrics: (1) the flood hazard, which is the intensity and frequency of a flood event; (2) the exposure, which are the
assets and/or people susceptible to flooding; and (3) the vulnerability, which are the economic and/or social consequences of
exposed elements as a result of a flood hazard (e.g., Klijn et al., 2015; Koks et al., 2015). The risk associated with flooding can
be diminished through the development of resilient infrastructure (e.g., Jongman, 2018; Woodward et al., 2014).

To develop such infrastructure under unknown future scenarios, quantifying the risk from compound flooding requires an
accurate quantification from the interactions between the many flood drivers impacting the flood hazard (e.g., Bates et al., 2023;
Woodward et al., 2013; Barnard et al., 2019). Currently, state-of-the-art stochastic compound flood risk assessments usually
apply the following four steps to quantify risk (e.g., Wyncoll and Gouldby, 2013; Couasnon et al., 2022; Rueda et al., 2015).
Firstly, the joint probability distribution between the selected flood drivers is modeled based on the observed dependence and
is used to generate synthetic events. Secondly, the flood hazard is modeled using a hydrodynamic model to account for the
non-linear interactions of the flood drivers. Thirdly, the damage is modeled by combining the flood hazard with information on
exposure and vulnerability. Lastly, the risk is modeled by accounting for the probability and damages associated with the flood
hazard. To obtain an accurate probability distribution of damages, many events must be generated in the first step. To this end,
a brute force Monte Carlo Simulation (MCS) of the selected flood drivers can be applied (e.g., Wu et al., 2021; Winter et al.,
2020). This can model the joint probability distribution of the different stochastic variables that define the time series of the
flood drivers. While this minimizes the number of simplifications, it requires a large number of simulations from hydrodynamic
models to quantify the flood hazard of each event, which can be computationally infeasible (e.g., Eilander et al., 2023b; Rueda
et al., 2015).

Consequently, compound flood risk assessments have focused on reducing the computational time of performing hydrody-
namic simulations while ensuring the risk estimate is accurate. Examples of strategies in the literature include the following:
(1) improving the computational resources (e.g., Apel et al., 2016); (2) using faster reduced-physics hydrodynamic models
(e.g., Bates et al., 2010; Leijnse et al., 2021); (3) reducing the number of hydrodynamic simulations through various sampling
techniques (e.g., Moftakhari et al., 2019; Barnard et al., 2019; Diermanse et al., 2014; Bakker et al., 2022); and (4) replacing
hydrodynamic simulations with data-driven (i.e., regression) models (e.g., Moradian et al., 2024; Fraehr et al., 2024). The
above examples are not mutually exclusive (e.g., Eilander et al., 2023b; Gouldby et al., 2017). Nonetheless, the largest reduc-
tion in computational time can be expected by focusing on the last two strategies (e.g., Rueda et al., 2015), which generate a
surrogate model by combining sampling and regression techniques to obtain an estimate of the damages from all events in the
MCS.

State-of-the-art surrogate models select simulations a priori by performing equidistant/factorial sampling for the events
contained in the MCS (e.g., Jane et al., 2022; Eilander et al., 2023b; Gouldby et al., 2017; Rueda et al., 2015). However,



compound flood events are characterized by many stochastic variables, such as the flood driver magnitude and duration, and the time lag between drivers. Therefore, a priori sampling requires a large number of hydrodynamic and impact simulations to provide a robust quantification for the damages of non-simulated events in the MCS. This often results in simplifications (e.g., Diermanse et al., 2023; Eilander et al., 2023b; Couasnon et al., 2022; Jane et al., 2022; Rueda et al., 2015). Examples of these are: (1) flood drivers are omitted; (2) stochastic variables such as the duration and the time lag are taken as constants; and (3) interpolation techniques are used for regression, as they minimize the computational time involved in training a data-driven model, which can be computationally expensive to ensure it generalizes well to unseen locations or forcing conditions (e.g., Fraehr et al., 2024; Moradian et al., 2024). These series of simplifications may impact the distribution of flood damages and result in a mis-quantification of compound flood risk.

This mis-quantification can be minimized by accelerating the process of obtaining a robust surrogate model. This can be potentially achieved by using active learning, which guides the sampling technique by optimizing for a goal. In the context of compound flood risk, a suitable goal could be to minimize the uncertainty related to the damages from the previous hydrodynamic and impact simulations. This would allow for hydrodynamic and impact simulations to be selected a posteriori. Treed Gaussian Process (TGP) models can make use of active learning and have been shown to reduce the number of hydrodynamic simulations associated with high dimensional datasets (e.g., Hendrickx et al., 2023, did so for modeling salt intrusion). Moreover, TGP models can also provide a reasonable regression model with limited training data (Gramacy and Lee, 2009). To the authors' knowledge, active learning is not used in state-of-the-art stochastic compound flood risk frameworks, but has successfully been used in other types of risk assessments (Tomar and Burton, 2021).

Therefore, this study aims to explore active learning to improve the quantification of compound flood risk assessments while limiting the increase in computational time. To this end, a new conceptual framework based on the TGP model is proposed, which (1) leverages the uncertainty in the response of damages to flood drivers to minimize the number of required hydrodynamic and impact simulations; and (2) can account for more stochastic variables in compound flood risk assessments. Therefore, this framework results in a more robust and comprehensive characterization of compound flood risk. As a proof of concept, the framework is applied to the case study of Charleston County in South Carolina (USA).

## 2 Methods

Our framework for compound flood risk assessments that utilize active learning uses the five following general steps, which can be visualized in (Fig. 1, e.g., Eilander et al., 2023b; Rueda et al., 2015):

1. Based on the characteristics of the case study, parameterize the selected flood drivers (Sect. 2.1).

2. Infer the natural variability of compound flood drivers to generate stochastic event sets (Sect. 2.2).

3. Simulate the damages associated with a synthetic event (Sect. 2.3).

4. Use a surrogate model to select simulations (synthetic events) with active learning and model the input-to-output (i.e., flood driver parameters to damages) relationship associated with a stochastic event set (Sect. 2.4).





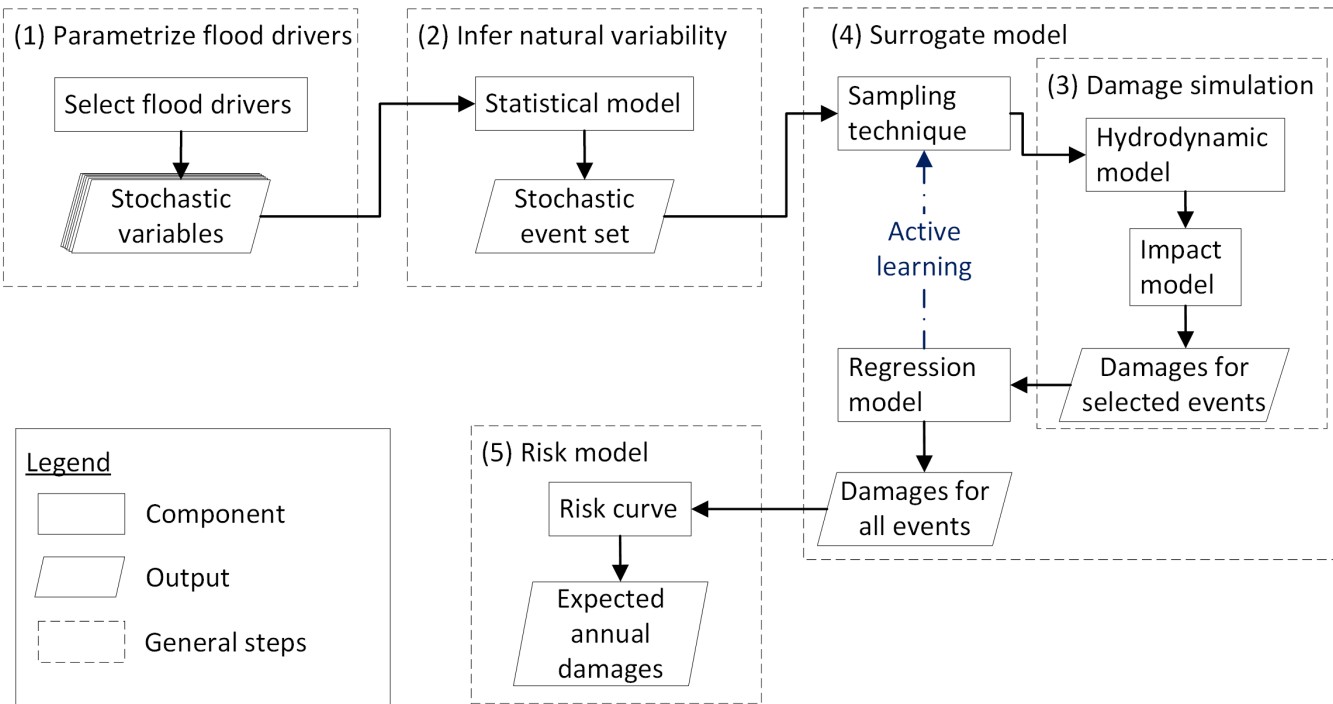

**Figure 1.** Typical framework required to characterize the flood risk associated with compound floods when using a surrogate model and our implementation for this study.

5. Model the risk by combining information on the probability and the damages of synthetic events to obtain an estimate of the Expected Annual Damages (EAD) (Sect. 2.5).

The main difference between the proposed framework and the current state-of-the-art lies in the fourth step (Sect. 2.4). Therefore, an experiment was designed in Sect. 2.6 to compare the active learning-based framework with a state-of-the-art one

95   based on equidistant sampling and a linear scatter interpolation (e.g., Eilander et al., 2023b; Jane et al., 2022).

## 2.1   Case study, flood drivers and data

Charleston County, located in South Carolina (USA), is on the coast of the Atlantic Ocean. Figure 2 shows Charleston County subdivided per each sub-county (United States Census Bureau, n.d.). According to their distance to the open coast, the sub-counties can be classed as either "inland" or "coastal" (Fig. 2).

100   Charleston County is prone to compound flooding caused by hurricanes and extra-tropical events (e.g., Parker et al., 2023; Nederhoff et al., 2024). These events cause precipitation, and wind shear and pressure effects offshore, which can result in the co-occurrence of astronomical tides, run-off, storm surges, and waves (e.g., Barnard et al., 2023), which regularly causes damages (e.g., Samadi and Lunt, 2023). Moreover, Sea Level Rise (SLR) is expected to worsen the flood hazards in the future





**Figure 2.** Charleston County with sub-county divisions which are color-coded according to their proximity to the open coast. Markers indicate the location of where data on the drivers is collected.

(e.g., Morris and Renken, 2020). This has caused the city to create a plan to manage its infrastructure (The City of Charleston, 105 2015).

Since validating a hydrodynamic model is not the purpose of this manuscript, the hydrodynamic SFINCS model (Leijnse et al., 2021) validated by Diermanse et al. (2023) for Charleston County was used (see Sect. 2.3). The small offshore model domain meant waves could not be taken into account (Diermanse et al., 2023). Moreover, waves are not a dominant driver for Charleston County (e.g., Parker et al., 2023). Thus, the same flood drivers as Diermanse et al. (2023) were investigated here. 110 These are the storm surge, tides, and precipitation.





Data for the still water level and the tides in Charleston Harbor were obtained at an hourly resolution from the National Oceanic and Atmospheric Administration (NOAA), at the tide gauge location shown in Fig. 2 (Station ID: 8665530). Data for precipitation was obtained at an hourly resolution from the ERA5 reanalysis dataset (Hersbach et al., 2020) at the grid location of 32.75º North, 79.75º West (see Fig. 2). The ERA5 dataset has a spatial resolution of 0.25º, roughly equivalent to 30 kilometers. The time series for the storm surge, tides, and precipitation have an overlapping time record of 24 years and 4 months.

To obtain the storm surge time series, the skew surge is considered as the difference between the highest still water level and the high tide within a tidal period (Williams et al., 2016; Couasnon et al., 2022; Diermanse et al., 2023). We further removed the sea level rise by subtracting a 1-year moving average from the skew surge time series (Arns et al., 2013). The latter was also used to identify the current sea level as 0.2 meters above Mean Sea Level (MSL).

Based on skew surge, tides, and precipitation, six stochastic variables were selected to parameterize compound flood events in Charleston County: skew surge magnitude ($S.Mag$); precipitation magnitude ($P.Mag$); tidal magnitude ($T.Mag$); precipitation duration ($P.Dur$); skew surge duration ($S.Dur$); and precipitation lag ($P.Lag$).

## 2.2 Inference of natural variability

Describing and inferring natural variability is key in risk management. Due to the relatively short time span of measured and/or reanalysis data, statistical models are often used to stochastically generate a large number of synthetic compound events (e.g., Couasnon et al., 2022; Bevacqua et al., 2017; Bates et al., 2023). The statistical model presented in the proposed framework follows the following four steps:

1. Identify extreme high-water events (Sect. 2.2.1).

2. Quantify flood driver parameters (Sect. 2.2.2).

3. Model the joint probability distribution (Sect. 2.2.3).

4. Generate stochastic event sets (Sect. 2.2.4).

### 2.2.1 Identify extreme high-water events

Peak Over Threshold (POT) was applied to $S.Mag$ when it co-occurred with a Higher High (HH) $T.Mag$. This allowed for the identification of extreme high-water events. POT identifies extremes as exceedances over a threshold, and uses a declustering time window to ensure all extremes are independent and identically distributed. Therefore, the number of identified extremes is dependent on the threshold and declustering time window. A threshold of 0.32 meters relative to MSL and a declustering time window of fourteen days between each extreme $S.Mag$ were chosen. This resulted in 2.91 extreme high-water events per year.





### 2.2.2 Quantify flood driver parameters

All six flood driver parameters had to be quantified for each extreme high-water event. POT was applied on $S.Mag$ in Sect. 2.2.1. For $P.Mag$, the largest value that co-occurred within $\pm$ three days of all identified $S.Mag$ extremes were used. For $T.Mag$, the co-occurring HH tide with all identified $S.Mag$ extremes were used. The $P.Dur$ and $S.Dur$ were taken as the duration of the $P.Mag$ and $S.Mag$ above a critical value, with a minimal duration of six days. For precipitation and skew surge, the values used to define the duration were 0.3 millimeters per hour and 0.2 meters respectively. The $P.Lag$ was defined as the difference in hours between the $S.Mag$ and $P.Mag$ for each extreme high-water event.

### 2.2.3 Model the joint probability distribution

The joint probability distribution between the different stochastic variables was modeled using a vine copula because of their flexibility in high dimensional datasets (Czado and Nagler, 2022; Bedford and Cooke, 2002) and their successful applications in other compound flooding studies (e.g., Bevacqua et al., 2017; Eilander et al., 2023b). A vine copula is defined by three components (e.g., Czado, 2019): (1) bivariate copulas; (2) a graph, named regular vine, which is composed by a series of nested trees; and (3) marginal Cumulative Distribution Functions (CDFs). A vine copula constructs a multivariate distribution using bivariate copulas. A bivariate copula models the dependence between two stochastic variables in the normalized ranked space. A vine copula organizes the bivariate copulas into a series of trees. The first tree represents the unconditional dependence between the stochastic variables. The following trees add a layer of conditional dependence. To transform the observations and generated data to and from the normalized ranked space where the vine copula is defined, marginal CDFs are required.

To select and fit a regular vine for a given problem, two options are possible: (1) brute force (Morales-Nápoles et al., 2023), or (2) heuristic algorithms. It has been shown that the number of regular vines grows extremely fast with the number of stochastic variables (Morales-Nápoles, 2010). Therefore, in this study the regular vine was chosen using Dißmann's algorithm (Dißmann et al., 2013) as implemented in the *pyvinecopulib* Python package (Nagler and Vatter, 2023) and fitted to minimize the Bayesian Information Criterion (BIC, Schwarz, 1978).

To mimic different levels of simplifications typically used in compound flood risk assessments, multiple (vine) copula models were fitted, each one considering different numbers of stochastic variables. The starting point was the copula model fitted between $S.Mag$ and $P.Mag$. Then, $T.Mag$, $P.Dur$, $S.Dur$, and $P.Lag$ were added one at a time. This resulted in one copula and four vine copulas fitted to two, three, four, five, and six stochastic variables, respectively. To simplify the models, if the stochastic variable that was added had independent copulas between all the pairs, it was removed from the vine copula. This simplified the number of models from five to three. Table A1 located in Appendix A shows an overview of the models.

Marginal CDFs were defined for each stochastic variable. For $S.Mag$, both the exponential and GPD were fitted to the data using the l-moments method and the best fit was selected based on the BIC using the *HydroMT* Python package (Eilander et al., 2023a). For $T.Mag$, the empirical CDF of all HH tides was used as the $T.Mag$ is expected to be independent of $S.Mag$ (e.g., Williams et al., 2016). For $P.Mag$, 80 continuous distributions available in the *scipy* Python package (Virtanen et al., 2020) were fitted using maximum likelihood estimates, and the best fit was selected based on the BIC. For the $P.Dur$, $S.Dur$, and





$P.Lag$, extrapolation is not desired, so only the truncated distributions available in the *scipy* Python package (Virtanen et al., 2020) were considered. The smallest Sum Squared Error (SSE) was used to choose the models. For $P.Mag$, $P.Dur$, $S.Dur$ and $P.Lag$ the *fitter* Python package was used (Cokelaer et al., 2024) to apply the exposed methodology. Table A2 located in Appendix A summarizes the CDFs and respective parameters chosen for the different stochastic variables.

### 2.2.4 Generate stochastic event sets

The fitted vine copula models were used to generate stochastic event sets to represent the different extents of simplifications in compound flood risk assessments. Stochastic event sets were generated with the inverse Rosenblatt transform in the *pyvinecopulib* Python package (Nagler and Vatter, 2023). When a variable was not yet stochastic, a constant was defined. This constant was assumed to be the median value from the empirical distribution. Moreover, in some cases, the incremental addition of a stochastic variable resulted in this variable only being contained in statistically independent bivariate copulas. To simplify, this variable was removed from the vine copula model, which resulted in the vine copula model only modeling statistically dependent variables. Thus, to obtain a stochastic event set that included a statistically independent variable, the stochastic event set generated by the simplified vine copula model was combined with data generated from the marginal of the independent variable.

Two different types of stochastic event sets were generated. For the first type, two "benchmark" event sets with 500 synthetic events with two and six stochastic variables were generated. For these event sets, the damage of each event was simulated (Sect. 2.3). For the second type, five "testing" event sets with 10,000 synthetic events with two, three, four, five, and six stochastic variables were generated.

## 2.3 Damage simulation

The damage related to synthetic events is required to train a surrogate model. This requires: (1) the simulation of the flood hazard; and (2) the simulation of the damages associated with a flood hazard. This study uses a hydrodynamic and impact model.

Many fully physics-solving and reduced physics hydrodynamic models are available (e.g., Delft-3D (Gerritsen et al., 2008); MIKE 21 (DHI, 2017); HEC-RAS (Brunner, 1996); LISFLOOD-FP (Bates et al., 2010)). The SFINCS (Leijnse et al., 2021) model was used to estimate the flood hazard map associated with the boundary conditions of a synthetic event. SFINCS was chosen because: (1) a model was validated for Charleston County; (2) the model uses high-resolution local datasets; and (3) the computational grid is small, reducing the computational time (Diermanse et al., 2023). SFINCS is a reduced-physics hydrodynamic model optimized for the fast calculation of the flood hazard (Leijnse et al., 2021). The governing equations are based on the local inertia equations (Bates et al., 2010), tuned for coastal and compound flooding by including additional terms, such as wind stress and advection. These equations are solved at the resolution of the computational grid using subgrid information about the topography and conveyance capacity (van Ormondt et al., 2024). For more details on SFINCS, see Leijnse et al. (2021) and van Ormondt et al. (2024). For our application, only two boundary conditions were required: (1) the still water level at the coast, and (2) the precipitation. The time series for the still water level was reconstructed by linearly





superposing three components: the constant MSL equal to 0.2 meters (Sect. 2.1); a historical tidal time series from the HH tide empirical distribution associated with a given $T.Mag$; and the skew surge time series. A Gaussian distribution was used to reconstruct the time series for skew surge and precipitation. For skew surge, $S.Mag$ and $S.Dur$ were used. For precipitation, $P.Mag$, $P.Dur$, and $P.Lag$ were used. Figure A1 in Appendix A shows how these variables were combined to create the

time series of boundary conditions for the downstream water level and precipitation. In terms of spatial distribution, both the downstream water level and precipitation were spatially uniform.

Impact models combine hazard, exposure, and vulnerability to quantify the economic and/or social consequences of a flooding event (e.g., Bates et al., 2023). The Delft-FIAT (Deltares, 2024) model was used to compute the damages associated with a synthetic event. Delft-FIAT was chosen because: (1) a model was validated for Charleston County, and (2) it uses data from

the United States Army Corps of Engineers and Federal Emergency Management Agency (Diermanse et al., 2023). Delft-FIAT combines the hazard map obtained from the SFINCS model, with the exposure (maximum damages of a building footprint), and the vulnerability (depth-damage fraction curves for each building footprint). This allowed for the computation of the damages associated with each building footprint included in the model for a synthetic event. Summing all building footprints in the model results in economic damage for Charleston County. Here, the damages associated with different geographic locations

(or outputs) were also investigated. This was done per sub-county and according to the classifications (i.e., coastal vs. inland) made in Sect. 2.1. These different spatial scales are referred to as the complete, sub-county, and classified models, respectively.

### 2.4    Surrogate model

A surrogate model approximates the behavior of a more complex and computationally expensive model. By being computationally faster, they are an asset in compound flood risk assessments for various reasons (e.g., quantifying uncertainty and

testing risk reduction measures (e.g., Eilander et al., 2023b)). The development of a surrogate model requires: (1) a sampling technique that selects a subset of events and configurations out of a large multivariate space; (2) running this subset in the reference numerical model; and (3) training a regression model to obtain the outcome for any other event or configuration possible. Previous compound flood risk studies only use surrogate models that: (1) sample based on input parameters (e.g, Jane et al., 2022), and (2) commonly combine with linear interpolation (e.g., Couasnon et al., 2022), although more complex regression

techniques are available (e.g., neural networks (e.g., Hendrickx et al., 2023); and radial basis functions (e.g., Antolínez et al., 2019)). However, active learning can be used to minimize the number of simulations by leveraging the uncertainty in the output (e.g., Tomar and Burton, 2021; Hendrickx et al., 2023). Here, the same sampling technique as in Hendrickx et al. (2023) was applied, which builds upon the work of Gramacy and Lee (2009). The tool used to perform active learning was also used as a regression model. Sect. 2.6 shows how these surrogate models were used and compared.

A Treed Gaussian Process, Limiting Linear Model (TGP-LLM; Gramacy and Lee, 2009) was used. It is a generative model able to provide different estimates for the damages. These estimates are used to calculate a mean and a confidence interval for the damages from each possible synthetic event from a stochastic event set. The mean is used as the regression model (Gramacy and Lee, 2009), while the confidence interval can be used as a metric to drive a sampling technique (e.g., Hendrickx et al., 2023). This is done by choosing the simulation with the largest standard deviation as it is expected to bring the largest





gain in information (MacKay, 1992). From here on, the standard deviation will be referred to as the Active Learning Mackay (ALM; MacKay, 1992) statistic.

A TGP-LLM uses Bayesian tree regression to partition the input into different subdomains, allowing different Gaussian Processes or linear models to fit different regions in the input space. This allows the TGP-LLM to be non-stationary and account for heteroskedasticity (Gramacy and Lee, 2009), preventing the magnitude of the ALM statistic from propagating

from one subdomain to another. This could result in large ALM statistics in uninteresting areas (Gramacy and Lee, 2009). Therefore, a TGP-LLM enables the selection of simulations to only occur in feature-rich sub-domains.

To minimize the number of simulations, performing active learning with the TGP-LLM model is beneficial. The conceptual framework, therefore, repeats three steps: (1) fit a TGP-LLM to the damages associated with the subset; (2) based on the highest ALM statistic, select a simulation to perform from the stochastic event set; and (3) perform the simulation to obtain the

damages, and append to the subset. The TGP-LLM has a computational cost that is proportional to the number of simulations currently in the subset ($N$): $O(N^3)$ (Gramacy and Lee, 2009).

When provided with a small subset of simulations, the TGP-LLM may sample randomly providing a small improvement in information for the computational cost (Hendrickx et al., 2023). To this end, we initially used a Maximum Dissimilarity Algorithm (MDA, Kennard and Stone, 1969) as it selects simulations at the outskirts of a stochastic event set (Camus et al.,

2011). Given an initial subset of simulated events, and a dissimilarity measure, a MDA repeats the following two steps until the final subset contains a predetermined number of simulated events. Firstly, given the current subset of simulated events, it assigns non-simulated events to the simulated event it is the least dissimilar to. Secondly, based on this assignment, the non-simulated event with the largest remaining dissimilarity to a simulated event is added to the subset and is simulated.

For our implementation, to prevent bias towards a variable, the stochastic event set was first normalized using a min-max

scaler from the *scikit-learn* Python package (Pedregosa et al., 2011). The MDA was initialized by providing the synthetic event associated with the largest $S.Mag$. Euclidean distance was used as a metric of dissimilarity. We used the MDA until $2^d$ (where $d$ represents the number of dimensions) events were contained in the subset, as it was proportional to the number of vertices. After the MDA simulations, active learning with the TGP-LLM was performed, as proposed by Hendrickx et al. (2023).

To further minimize the computational cost, we developed our own stopping criterion as this is still an active area of research

(e.g., Ishibashi and Hino, 2021; Tomar and Burton, 2021; Hendrickx et al., 2023). The learning curve of the TGP-LLM was assessed by comparing its mean with the benchmark (Sect. 2.2.4) after each TGP-LLM iteration. To this end, different metrics were computed: the Root Mean Square Error (RMSE) of the simulated and non-simulated events, the mean and maximum ALM statistic of the non-simulated events, and the Expected Annual Damages (EAD) (Sect. 2.5). A two-sample Kolmogorov-Smirnov (KS) test (Sect. 2.6) was applied to compare the empirical CDFs of the benchmark and the TGP-LLM.

The stopping criterion was defined as an ALM mean smaller than 0.1 for two consecutive TGP-LLM models for an output. This is based on our findings that: (1) the mean ALM is correlated with the RMSE of the simulated events; (2) the EAD shows more stability and certainty as the number of simulations increases; (3) the change in the RMSE per simulation is below 1 million USD per simulation when the mean ALM is below 0.1; and (4) the two-sample KS test always has a significant $p$-value for six stochastic variables (Fig. A2).





The implementation of a stopping criterion meant it was unknown how well these simulations would perform for an unseen output. Therefore, a round-robin schedule was used. After each simulation, the TGP-LLM was fit to a different output in a predetermined loop. This process was repeated until all outputs for a model reached the stopping criterion. To reduce the computational costs, outputs that reached the stopping criterion were incrementally removed from the round-robin schedule.

## 2.5    Risk modelling

To model the risk, the method used by Couasnon et al. (2022) was followed. Each event in the stochastic event set was assumed to occur at a constant frequency, equivalent to the reciprocal of the average number of floods per year identified by the POT. Then, all events were ranked according to the magnitude of their economic damage, creating an empirical CDF. The risk curve was obtained by converting the rank of each event to a Return Period (RP, Gumbel, 1941). The EAD is an important metric in flood risk assessments (e.g., Olsen et al., 2015) as it allows the performance of a cost-benefit analysis (e.g., Haer et al., 2017).
To obtain the EAD, the empirical CDF of damages was integrated.

## 2.6    Experiment

An experiment was designed to validate, test, and assess the proposed framework under different scenarios. Firstly, a comparison was made with the state-of-the-art (Sect. 2.6.1). Secondly, the scalability was tested (Sect. 2.6.2). Finally, the effect of simplifications on flood risk was assessed (Sect. 2.6.3).
To compare the different risk estimates, statistical tests were performed. A two-sample Kolmogorov-Smirnov (KS) test was applied to compare two empirical CDFs (Hodges, 1958). The null hypothesis assumes that both empirical CDFs are drawn from the same parent distribution. To compare two EADs, a Mann-Whitney U (MWU) Rank test was used (Mann and Whitney, 1947). The null hypothesis assumes both empirical CDFs have the same EAD. For the latter, an empirical bootstrap (Efron, 1979) was repeated 500 times for each empirical CDF to obtain a distribution of EADs. For the implementation of both tests,
the *scipy* Python package (Virtanen et al., 2020) was used. Two risk estimates were considered significantly different if the $p$-value was smaller than 0.05.

When comparing the computational cost of different surrogate models, the computational time was assessed on an AMD Ryzen 5 5600X 6-Core Processor 3.70 GHz CPU.

### 2.6.1    State-of-the-art vs. active learning

To validate the proposed framework, the active learning approach was compared with a current state-of-the-art equidistant sampling approach based on the methods used by Jane et al. (2022) and Eilander et al. (2023b). Both approaches were only compared for the models with two stochastic variables.

For the current state-of-the-art approach, MDA sampling was combined with linear scatter interpolation based on the recommendation of Jane et al. (2022). In a linear scatter interpolation, simulated events are triangulated (for two dimensions),
and linear interpolation occurs within each triangle. To prevent extrapolation, $2^2$ simulations representing the vertices of the





stochastic event set were simulated. Then, a MDA was applied to the stochastic event set using the same implementation as the active learning approach (Sect. 2.4). In total 64 events were simulated, which is comparable to the number of events used by Eilander et al. (2023b) based on factorial sampling with $8^d$ simulations. The *gridddata* function from the *scipy* Python package was used to implement the linear scatter interpolation (Virtanen et al., 2020).

The testing and benchmark event sets in two dimensions were used to compare both approaches. The testing event set was used to select the simulations to run as it provided a larger diversity in the synthetic events. Then, the regression model was used to calculate the damages of all synthetic events in the benchmark event set, which allowed for the quantification of the RMSE.

### 2.6.2  Scalability of the proposed framework

To test the scalability of the proposed framework, the active learning approach was deployed on the testing event sets. Since compound flood drivers respond differently based on the geographic location (e.g., Gori et al., 2020), the active learning approach was not only deployed on the complete (all sub-counties combined) model but also on the classified (coastal and inland sub-counties separated) model for all testing events. It was also deployed on the testing event for two stochastic variables for the sub-county model.

### 2.6.3  Effect of simplifications on flood risk

To assess the effect of simplifications on current compound flood risk assessments, the flood risk associated with the damage of the complete model to the five testing event sets was modeled.

## 3  Results and discussion

### 3.1  State-of-the-art vs. active learning

This section presents and discusses the accuracy and computational time of equidistant sampling and active learning surrogate models. Figure 3 shows the RMSE of both surrogate models compared to the benchmark events set. The active learning approach outperforms the equidistant sampling approach in two ways: (1) for any number of simulations, the RMSE is smaller; and (2) the smallest RMSE is reached after fewer simulations (14 vs. 52). This results in the active learning approach improving the accuracy by a factor of eight (90.8 vs. 11.2 million USD), while reducing the number of numerical simulations that would
normally be performed by a factor of four (64 vs. 14). This increase in accuracy provides a better estimate of the EAD for the active learning approach (37.3% vs. 1.68% error) when compared to the benchmark EAD (Fig. A3).

The difference between the accuracy and number of numerical simulations of both approaches is caused by their respective sampling and regression techniques. On the one hand, equidistant sampling cannot explore uncertainty and thus selects a large number of numerical simulations. When combined with linear scatter interpolation, it is not flexible enough to correctly
represent the non-linear response of damages to the flood drivers. To achieve a similar accuracy to the active learning approach,


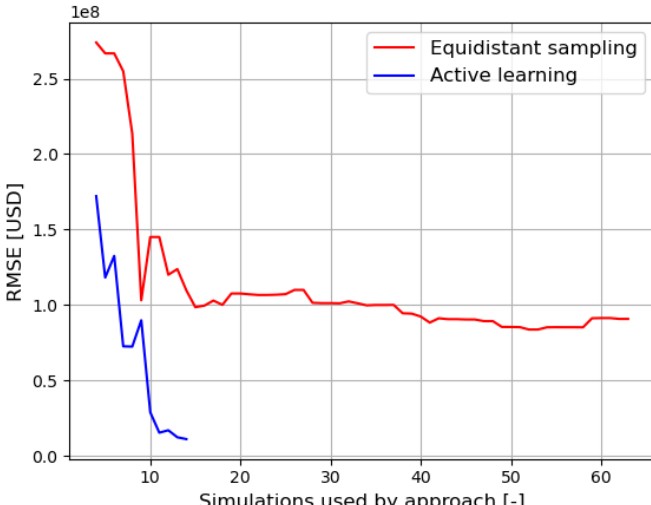

**Figure 3.** RMSE of both approaches as a function of the number of simulations from the testing event set with two stochastic variables, measured based on the difference with benchmark events.

a large number of numerical simulations would be required, increasing the computational cost. This is similar to observations made by Gouldby et al. (2017). If the number of stochastic variables were to increase, the curse of dimensionality would cause the number of simulations to increase exponentially, making the problem infeasible. On the other hand, the active learning approach explores the uncertainty related to the output (i.e., damages related to a geographic location) of simulations already

performed. This allows for the sampling of simulations that are expected to bring the largest gain in expected information. This is combined with a generative model that can capture some of the non-linear relationships between the flood driver parameters and the damages. However, the active learning approach is unable to provide a perfect fit as the RMSE never reaches 0 (Fig. A2), showing that a stopping criterion is necessary, as marginal gains in accuracy may bring high computational costs. This causes the active learning approach to show significant differences with the benchmark when using the two-sample KS test

(Fig. A3).

Figure 4 shows the computational time associated with both approaches when they sample from the testing event set. The computational time can be split into three components: (1) Delft-FIAT, (2) SFINCS, and (3) TGP-LLM. For both approaches, Delft-FIAT is the component that requires the largest computational time. This is caused by the Delft-FIAT model being poorly optimized for Charleston County as it preprocesses the exposure data before each simulation. The added computational

time for the active learning approach due to the TGP-LLM is relatively small as the number of simulations is small (2.3 minutes). Therefore, the number of simulations is the main factor influencing the computational time. For our experiment, the computational time was reduced by a factor of four (95.4 vs. 23.6 minutes).





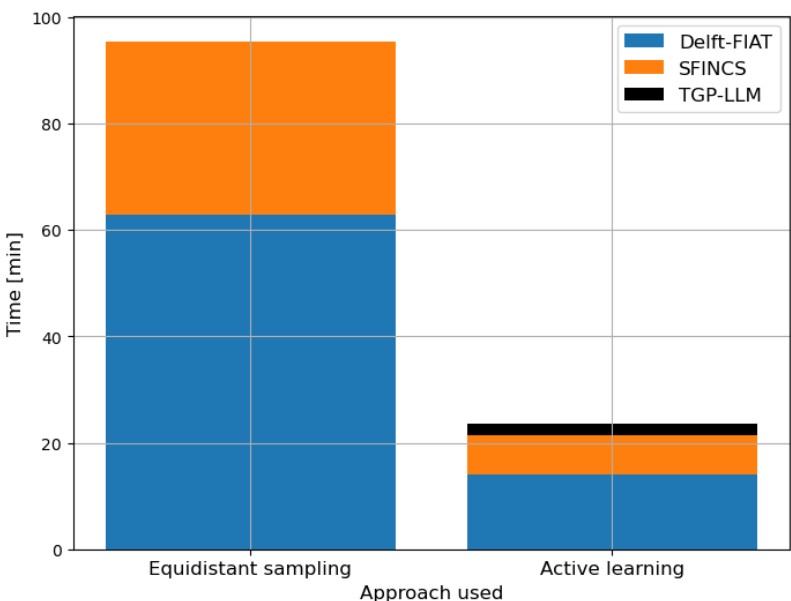

**Figure 4.** Computational time associated with both approaches for the testing event set with two stochastic variables.

## 3.2 Scalability of the proposed framework

In this section, we present and discuss the computational time of the active learning approach under different extents of sim-

plifications. These simplifications can take various forms, but we only investigate two of these: (1) the inclusion of additional

stochastic variables; and (2) increasing the number of outputs. Figure 5 shows the computational time of the active learning

approach for the testing event sets with different numbers of stochastic variables (or dimensions) and the number of outputs.

For each, the computational time is subdivided into three components: (1) Delft-FIAT, (2) SFINCS, and (3) TGP-LLM. A

horizontal red line is provided as a reference, showing the computational time of the equidistant sampling approach with two

dimensions. For a singular output (complete), the reference computational time is only exceeded with six dimensions. The dif-

ference in computational time between two and six dimensions is a factor of five (100 minutes). The computational time for two

outputs (classified), is larger for all test event sets when compared to a singular output (complete). For one and two outputs, the

computational time does not always increase as the number of dimensions increases. Moreover, the TGP-LLM dominates the

computational time of the active learning approach for the classified model in six dimensions, where the TGP-LLM amounts

to 75% (1163.9 out of 1553.3 minutes) of the computational time. This caused it to exceed the reference computational time

by a factor of sixteen. Finally, when increasing the number of outputs to eleven (sub-county), the active learning approach is

below the reference computational time but requires substantially more computational time compared to one and two outputs.

    For a singular output, the cost associated with the TGP-LLM is limited as the stopping criterion is often met quickly after

the MDA initialization (Table A3). Thus, the TGP-LLM only has to select a small number of simulations, which dominates


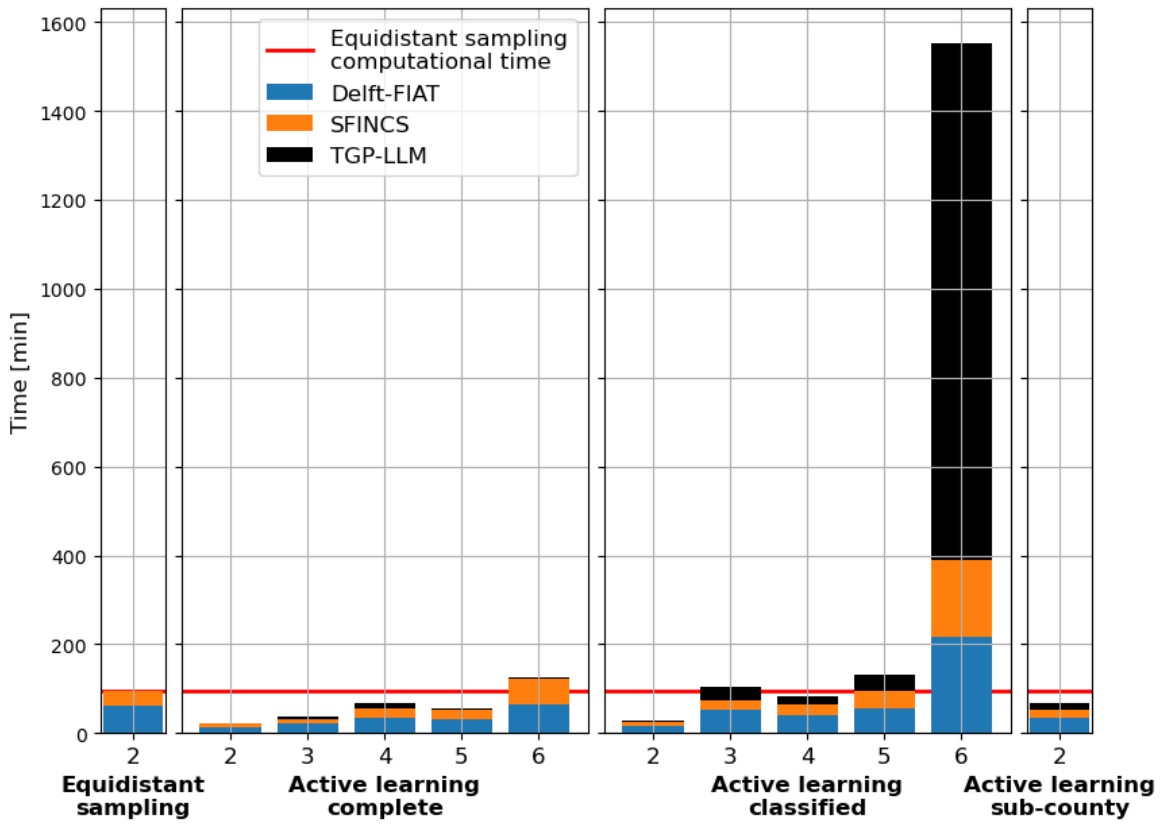

**Figure 5.** Computational time required for different approaches as a function of outputs (different geographic locations) and number of stochastic variables (dimensions).

the reduction in the computational time. The use of an initialization results in five dimensions having a smaller computational time than four dimensions. This is because the TGP-LLM was used a smaller number of times.

When increasing the number of outputs to two, the number of simulations also increases. This is because the response of economic damages to the flood drivers is different for both locations. This can be seen in Fig. A4 and is a known phenomenon in compound flood risk (e.g., Gori et al., 2020). The number of additional simulations will depend on how complex and diverse

the response surfaces are. Here, the more complex response surface of the inland location requires a larger number of numerical simulations than the coastal location (Table A3). These additional simulations cause the computational time for the classified model to increase when compared to the complete model. This increase in simulations also causes the relative cost of the TGP-LLM to increase as its cost is proportional to the number of simulations cubed (Gramacy and Lee, 2009).





For eleven outputs, the computational time significantly increases when compared to one and two outputs. This is not only
caused by the response of the flood drivers to each location but also by the stopping criterion. The round-robin schedule requires
a minimum number of simulations, which is proportional to the number of outputs (Sect. 2.4).

### 3.3 Effect of simplifications on flood risk

This section presents and discusses the outcomes of incrementally adding stochastic variables to the estimate of economic
risk. This is tested for a singular output using the active learning approach. Figure 6 shows the risk curves associated with the
different numbers of stochastic variables (or dimensions). The samples represent the mean response of the TGP-LLM to a test
event set in $d$ dimensions. Additionally, since the TGP-LLM is a generative model (i.e., it captures and models the distribution
of the output), the uncertainty associated with the surrogate damage model at each RP can be computed by modeling the risk
associated with the 5% and 95% confidence interval in damages for each event. Figure 6 shows that these uncertainty bands are
independent of the number of dimensions. The EAD associated with each risk curve is also shown. The logarithmic behavior
of the x-axis makes it easier to see the differences in the economic damages at large RPs. When considering the uncertainty
of the TGP-LLM, the model with two dimensions consistently underestimates the economic damages for RP > 10 years when
compared to models in higher dimensions. In addition, based on the two-sample KS test, all combinations show significant
differences (Fig. A5). When considering the EADs, all but one of the combinations show a significant difference (Fig. A6).
When comparing models with two and three dimensions (all driver magnitudes) and the three and six dimensions (adding driver
durations and time lag), the EAD difference is 11.3% (22.01 million USD) and 11.6% (25.47 million USD), respectively. The
return values are also directly compared. For a RP of 1, 10, and 100 years the differences between: (1) two and three dimension
models are 2.98, 115.08, and 279.57 million USD respectively; and (2) six and three dimension models are 3.03, 31.03, and
66.08 million USD respectively.

The uncertainty bands are independent of the number of dimensions of the model, as the stopping criterion is based on an
uncertainty-based threshold. The uncertainty can be reduced by using a stricter stopping criterion, providing a larger confidence
in the risk curve. However, this will come at the expense of a larger computational cost.

The main reason for the risk curve in two dimensions showing large differences with other risk curves in higher dimensions
is the omission of the $T.Mag$ as a stochastic variable. This greatly influences the quantification of flood risk as Charleston
County is located close to the open coast. This means that the tide and the surge drivers will have a large effect on the economic
damage. When $T.Mag$ is included, the more extreme coastal water levels have a larger magnitude. Combined with non-linear
vulnerability curves (Diermanse et al., 2023), this causes a non-linear increase in damages.

Significant differences between almost all risk curves and EAD shows that neglecting drivers' duration and time lag will
lead to a mis-quantification in flood risk estimates. In general, the effect of an added stochastic variable on the flood risk
depends on two things: (1) how the economic damages respond to the stochastic variable; and (2) the assumed constant value
used in a model with fewer dimensions and how well that represents its probability distribution and dependence with other
stochastic variables. If a constant is used to represent a stochastic variable, it will be unable to provide the same risk curve but
may provide a similar estimate in the EAD or return value. For the EAD, this is shown in Fig. A5 and Fig. A6 where two and


**Figure 6.** Risk curves for models with a different number of stochastic variables (dimensions). The uncertainty bands represent the 5th and 95th percentiles. The legend includes the EAD estimate for each model.

four dimensions show significantly different risk curves, but have similar estimates in the EAD. This is because the EAD is an integral of the empirical CDF. Thus, differences can be offset, showing the EAD isn't the best metric to use when justifying

simplifications.



## 3.4 Limitations

In this study, the surrogate models were assessed for the case study of Charleston County. Other case study locations will show a different response to economic damages as these will: (1) have a different combination of flood drivers (e.g., Eilander et al., 2023b); (2) have different physical properties that determine the response of the flood hazard to the drivers; and (3) have different spatial distributions of exposure and vulnerability (e.g., Koks et al., 2015). While the findings about the exact reduction in computational time and increased accuracy are case-specific, we expect similar results in other locations. Active learning allows for the number of numerical simulations to be minimized given an input-to-output relationship. The number of simulations will depend on how complex this relationship is, but as shown in Sect. 3.2, the TGP-LLM is still able to reach the stopping criterion for different outputs that have a different response to the flood drivers. The results of this study only investigate up to six stochastic variables, but certain case study locations can be affected by more than six drivers (e.g., California Bay-Delta, Cloern et al., 2011), leading to a stronger curse of dimensionality if their duration, time lag, and spatial distribution are also included. We expect that the conceptual framework can still be applied to these case studies, but the computational time will largely depend on the interaction of the flood drivers and, the number of outputs to be modeled. The computational time of the conceptual framework can be further minimized by reducing the number of times the TGP-LLM is applied (e.g., each $x$ simulation rather than after each single simulation).

Choosing the number of flood drivers and associated stochastic variables to include is currently based on knowledge from previous studies, expert knowledge, or a preliminary sensitivity analysis. We expect the TGP-LLM to understand if a stochastic variable is not contributing to economic damages as it can associate a linear model with this dimension. However, this will require additional numerical simulations. Moreover, it may become difficult to generate representative synthetic events from a robust statistical model in higher dimensions (Morales-Nápoles et al., 2023).

SFINCS is used here as a hydrodynamic model as it has been validated during a previous study (Diermanse et al., 2023). SFINCS significantly decreases the computational cost associated with the hydrodynamic simulation of an event. However, it neglects certain physical processes (e.g., morphodynamics) (Leijnse et al., 2021), potentially increasing the modeling uncertainty associated with the flood hazard. The use of the conceptual framework proposed in this study reduces the number of numerical simulations. This allows for the use of more computationally demanding numerical models, which are expected to represent the flood hazard better in more complex cases. Nonetheless, the majority of the modeling uncertainty is caused by the input datasets used to calibrate and validate the hydrodynamic model, which would also be present in more complex hydrodynamic models (Bates et al., 2021).

When fitting the statistical model to the observed extreme high-water events, we grouped extra-tropical and tropical events because of the short time record for the available data (see Sect. 2.1). Ideally, these should be taken into account using separate distributions when modeling risk (e.g., Nederhoff et al., 2024).

During this study, only the uncertainty associated with economic consequences was explored. However, risk can also be social. Authors who use the equidistant/factorial sampling approach make a necessary simplification that the simulations chosen to represent economic risk also represent social risk (e.g., Diermanse et al., 2023; Eilander et al., 2023b). This simplification





is not required for the active learning approach as the social consequences can be included as an additional output in the round-robin schedule of the TGP-LLM.

## 4   Conclusions and recommendations

A conceptual framework that uses active learning to leverage the input-to-output uncertainty was applied to the case study of Charleston County. The proposed framework uses uncertainty related to the economic damages caused by flood driver

parameters. This framework reduces the computational time of performing a compound flood risk assessment and/or allows for reducing the number of simplifications usually taken in compound flood risk assessments.

When comparing the state-of-the-art equidistant sampling surrogate model with the proposed active learning surrogate model, the RMSE in damages was reduced by a factor of eight (90.8 vs. 11.2 million USD), while reducing the computational time by a factor of four (95.4 vs. 23.6 minutes), resulting in a win-win scenario. This reduction in error results in higher

accuracy risk estimates. The reduction in computational time makes it possible to include more stochastic variables (i.e., to reduce the number of simplifications) to improve risk estimates. For a singular output (one geographic location), the computational time increased by a factor of five (100 minutes) between two and six stochastic variables. Not accounting for all the simplifications resulted in an underestimation of the Expected Annual Damages (EAD) of 47.48 million USD, and an underestimation of the 100-year Return Period (RP) of 345.65 million USD. Exploring the uncertainty associated with different

outputs (i.e., geographic locations) increased the computational time as the economic damages responded differently to the flood drivers, requiring additional numerical simulations. In the case of six dimensions and two outputs, this caused the computational time to increase by a factor of sixteen when compared with the current state-of-the-art. While the proposed active learning surrogate model was only assessed for a single case study, there is a high expectation to see similar benefits to other case studies.

Based on these findings, we have three recommendations for future studies. Firstly, different strategies could limit the computational time associated with training the Treed Gaussian Process, Limiting Linear Model (TGP-LLM) model. For instance, simulating the synthetic events with the $x$ highest Active Learning Mackay (ALM) statistic in each iteration or increasing the number of simulations in the initialization. Secondly, some case studies will have stochastic variables associated with processes that do not dominate the compound flood risk. To limit the increase in computational time, future research should propose sim-

plifications that do not affect the quantification of compound flood risk. However, our results show that similar EAD values can result from different risk curves, and hence, care should be taken when using EAD to validate these simplifications. Finally, to propose optimal risk reduction measures, the framework would have to be deployed once for each possible adaptation measure. Future research should investigate how to limit the computational cost of this operation by understanding and predicting how the input-to-output response surface will change with certain adaptation measures.




*Code and data availability.*    The scripts and data used to set up the experiments in this study are available from Zenodo at https://doi.org/10.5281/zenodo.13910108 (Terlinden-Ruhl, 2024).

## Appendix A:  Supplementary information

**Figure A1.** Schematization of SFINCS model boundary conditions.



**Figure A2.** Developing a stopping criterion for compound flood risk. **(a)**, **(b)**, **(c)**, **(d)** and **(e)**, **(f)**, **(g)**, **(h)** represent the benchmark datasets for two and six dimensions respectively. **(a)**, **(e)** shows the error of the surrogate damage model. **(b)**, **(f)** shows information on the ALM statistic. **(c)**, **(g)** compares the estimate of the EAD from the surrogate damage model with the benchmark. **(d)**, **(h)** shows the results of the two-sample KS test when comparing the empirical CDF of the surrogate damage model with the ground truth.


**Figure A3.** Risk estimates of both approaches when compared with the ground truth. **(a)**, **(b)** show the outcomes of the two-sample KS test for the active learning and equidistant sampling approaches respectively. **(c)** shows the estimates of EAD from each approach.




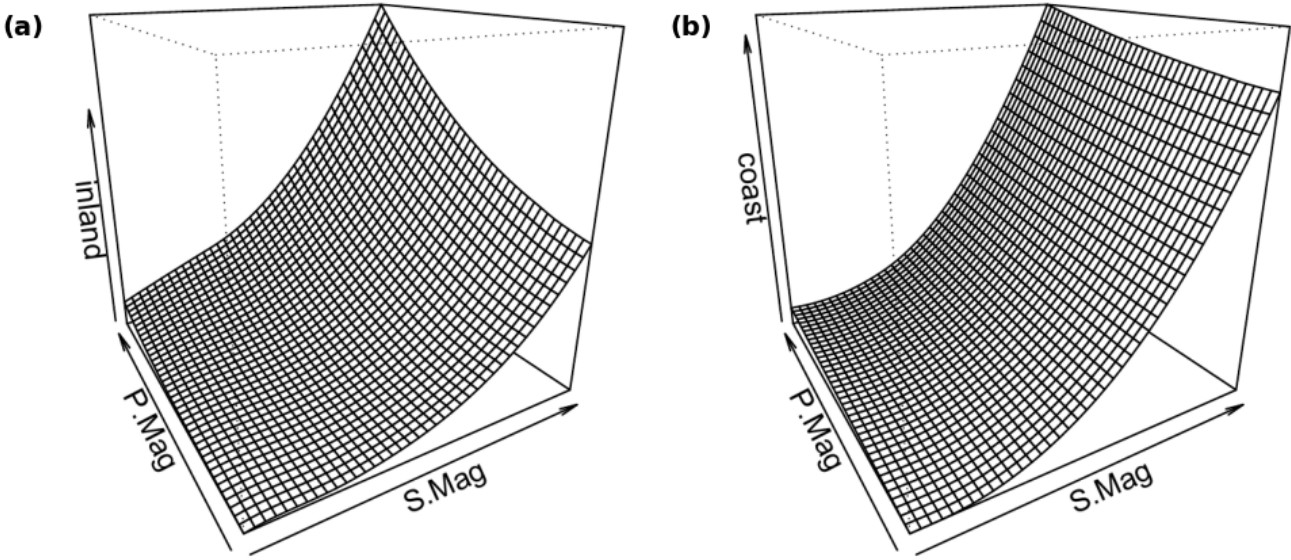

**Figure A4.** Response of economic damages to the testing event set in two dimensions for the different outputs of the classified model: **(a)** shows the response of the inland location; **(b)** shows the response of the coastal (coast) location. Plots are obtained from the tgpllm R package developed by Gramacy and Lee (2009).





**Figure A5.** Test statistics and associated $p$-values associated with two-sample KS test for the empirical CDFs of different dimensionalities when sampling from the total economic damages. If the $p$-value is smaller than 0.05, the null hypothesis (empirical CDFs come from the same parent distribution) is rejected.



**Figure A6.** P-values associated with MWU test for empirical CDFs of different dimensionalities. If the $p$-values are smaller than 0.05, the null hypothesis (empirical CDFs have the same location (EAD)) is rejected.





**Table A1.** Regular vine copula models (e.g., Czado, 2019) for different numbers of stochastic variables with associated regular vine distribution and bivariate copula models.

| 2 stochastic variables | | | |
|---|---|---|---|
| Tree | Edge | Copula model | Parameters |
| 1 | *S.Mag, P.Mag* | Gaussian | 0.348 |
| **4 stochastic variables** | | | |
| Tree | Edge | Copula model | Parameters |
| 1 | *S.Mag, P.Mag* | Gaussian | 0.348 |
|  | *P.Mag, P.Dur* | Joe 180$^o$ | 2.44 |
|  | *T.Mag, P.Dur* | Frank | 1.69 |
| 2 | *T.Mag, P.Mag ∣ P.Dur* | Independence | |
|  | *P.Dur, S.Mag ∣ P.Mag* | Independence | |
| 3 | *T.Mag, S.Mag ∣ P.Dur, P.Mag* | Independence | |
| **6 stochastic variables** | | | |
| Tree | Edge | Copula model | Parameters |
| 1 | *T.Mag, P.Dur* | Frank | 1.69 |
|  | *P.Dur, P.Mag* | Joe 180$^o$ | 2.44 |
|  | *P.Mag, S.Mag* | Gaussian | 0.348 |
|  | *S.Dur, S.Mag* | Independence | |
|  | *S.Mag, P.Lag* | Frank | 1.98 |
| 2 | *T.Mag, P Mag ∣ P.Dur* | Independence | |
|  | *P.Dur, S.Mag ∣ P.Mag* | Independence | |
|  | *P.Mag, S.Dur ∣ S.Mag* | Independence | |
|  | *S.Dur, P.Lag ∣ S.Mag* | Joe 270$^o$ | 1.35 |
| 3 | *T.Mag, S.Mag ∣ P.Dur, P.Mag* | Independence | |
|  | *P.Dur, S.Dur ∣ P.Mag, S.Mag* | Independence | |
|  | *P.Mag, P.Lag ∣ S.Mag, S.Dur* | Joe 180$^o$ | 1.30 |
| 4 | *T.Mag, S.Dur ∣ P.Dur, P.Mag, S.Mag* | Independence | |
|  | *P.Dur, P.Lag ∣ P.Mag, S.Mag, S.Dur* | Independence | |
| 5 | *T.Mag, P.Lag ∣ P.Dur, P.Mag, S.Mag, S.Dur* | Independence | |



**Table A2.** Summary of the marginal CDFs for the different stochastic variables.

| stochastic variable | Distribution | Parameters |
|---|---|---|
| $S.Mag$ | Exponential | {Mean: 0.322; Scale: 0.102} |
| $P.Mag$ | Exponential | {Mean: 0.0796; Scale: 5.60} |
| $T.Mag$ | Empirical: HH tides | |
| $P.Dur$ | Truncated Normal | {a: 0; b: 406; Mean: 0.158; Scale: 18.9} |
| $S.Dur$ | Truncated Gumbel | {c: 0.726; Mean: 0.880; Scale: 5.26} |
| $P.Lag$ | Truncated Normal | {a: -72.0; b: 72.0; Mean: -10.7.; Scale: 30.6} |



**Table A3.** Total number of simulations required to reach stopping criterion for different dimensions and number of outputs. For multiple outputs, these are ordered top-down in the order they appear in the round-robin schedule. **Bold** numbers represent the output dictating the total number of simulations required to reach the final stopping criterion.

| | | Number of dimensions [-] | | | | |
|---|---|---|---|---|---|---|
| Model | Output | 2 | 3 | 4 | 5 | 6 |
| Complete | Total | **14** | **24** | **36** | **33** | **65** |
| Classified | Inland | **17** | **38** | **43** | **59** | **224** |
| | Coastal | 13 | 30 | 37 | 35 | 67 |
| Sub-county | Charleston Central | **36** | | | | |
| | Edisto Island | 27 | | | | |
| | James Island | 28 | | | | |
| | Johns Island | 29 | | | | |
| | Kiawah Island | 30 | | | | |
| | Seabrook Island | | | | | |
| | McClellanville | 31 | | | | |
| | Mount Pleasant | 32 | | | | |
| | North Charleston | 33 | | | | |
| | Ravenel Hollywood | 34 | | | | |
| | Wadmalaw Island | 24 | | | | |
| | West Ashley | 35 | | | | |

*Author contributions.* Author contributions follow the CRediT authorship categories. L.TR: Conceptualization, Methodology, Software, Formal analysis, Investigation, Writing—original draft, Writing—review and editing and visualization. A.C, D.E, G.G.H: Conceptualization, Methodology, Writing—original draft, Writing—review and editing. P.MN: Writing—original draft, Writing—review and editing, J.A.A.A: Writing—original draft, Writing—review and editing, Supervision.

*Competing interests.* Anaïs Couasnon is a member of the editorial board of the special issue: Methodological innovations for the analysis and management of compound risk and multi-risk, including climate-related and geophysical hazards.

*Acknowledgements.* We would like to thank the FloodAdapt team for providing their support with the hydrodynamic and impact models. This research was funded by Deltares' Moonshot 2: Making the world safer from flooding.



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
