# Peer review of "Accelerating compound flood risk assessments through active learning: A case study of Charleston County (USA)"

_Natural Hazards and Earth System Sciences, 2024_

## Author Comment (AC1)

**Referee Comment 1**

The study explores the application of active learning to improve the accuracy of compound flood risk assessments while minimizing computational demands. The framework leverages the input-output uncertainty related to economic damages to minimize the number of required hydrodynamic and impact simulations. Compared to traditional equidistant sampling, the proposed active learning modeling framework shows lower RMSE with lower computational time.

I would like to compliment the authors for a well-written manuscript. To the best of my knowledge, this is the first study to apply active learning techniques in the context of compound flooding, demonstrating an approach to advancing this area of research. I recommend this manuscript for publication after addressing the minor revisions outlined below. However, I am not qualified to evaluate the detailed technical aspects of active learning.

The authors appreciate the positive and constructive comments provided by the first referee, which can only improve the quality of the manuscript.

1.  The authors used the "skew surge" parameter to represent the storm surge component of water levels. However, one of the key parameters for simulating flooding mentioned in the manuscript is the "duration" of the event. The skew surge is a measure of the storm surge integrated over a tidal cycle, and thus it has no duration associated with it. How are the authors defining the duration (or time series) of the skew surge? In addition, SFINCS needs a water level time series as boundary conditions to simulate coastal flooding, how do the authors generate a storm-tide hydrograph of these events using the skew surge (i.e. a single value over a tidal cycle)? Consider explaining more about these steps.

    The authors agree with the reviewer that the description of the skew surge was vague within the manuscript, which could lead into confusion with the quantification of the skew surge duration ($S.Dur$).

    As described in L 117-118, the skew surge is the difference between the highest still water level and high tide within a tidal period. This indeed results in a single value per tidal period. Because of the semi-diurnal tidal regime at the tidal gauge of Charleston Harbor, values for skew surge were measured every 12 hours. To define the skew surge time record, it was assumed the skew surge values were constant for the tidal period during which they were calculated. This causes a stepwise time record. Hence, the skew surge duration can increase in 12-hour increments from a starting value of 12 hours.

    The authors have made the following changes:

*L 117 " Increased still water levels from storm surges induce a phase shift in the tidal signal, creating spurious peaks in the storm surge time record (Williams et al., 2016). Therefore, the skew surge  was considered. It is the difference between the highest still water level and the high tide within a tidal period (Williams et al., 2016; Couasnon et al., 2022; Diermanse et al., 2023). A tidal period was taken as the time between two consecutive low tides. To recreate a time record with an hourly resolution, the skew surge values were assumed to be constant over their tidal period, which was approximately 12 hours for Charleston Harbor."*

With this clarification, the authors believe the current text will clearly describe the generation of the water level time series boundary condition for SFINCS. L 143-144 shows the $S.Dur$ was quantified in 12-hour increments from 12 hours to a maximum of 6 days (see answer to comment 9) because the skew surge time series was measured every 12 hours.

*L 143 "The P.Dur and S.Dur were taken as the duration of the P.Mag and S.Mag to continuously remain above a critical value within the ± three-day window used to quantify P.Mag. ."*

Furthermore, L 205-208 describes how $S.Dur$ and $S.Mag$ were combined to create a skew surge time series. The time series was then combined with a tidal time series and a Mean Sea Level (MSL) component to generate a water level time series for SFINCS.

*L 205 "The time series for the still water level was reconstructed by linearly superposing three components: the constant MSL equal to 0.2 meters (Sect. 2.1); a historical tidal time series from the HH tide empirical distribution associated with a given T.M ag; and the skew surge time series. A Gaussian distribution was used to reconstruct the time series for skew surge and precipitation. For skew surge, S.Mag and S.Dur were used."*

Figure A1 can also be used to visualize the process of generating boundary conditions for SFINCS with the six flood driver parameters.

[Figure]

**Figure A1.** Schematization of SFINCS model boundary conditions.

2. The methodology for obtaining the skew surge time series is somewhat unclear (see L 112). Specifically, the paper does not specify how tidal levels were determined. Were NOAA-predicted tides used, or was a harmonic analysis performed on water level data? Additionally, when removing sea-level rise, the phrase "subtracting 1-year moving average from the skew surge time series" is unclear.

To increase clarity in the methodology for obtaining the skew surge time recordt, the authors have made the following changes:

L 111 "Data for the still water level and the tides in Charleston Harbor were obtained at an hourly resolution from the National Oceanic and Atmospheric Administration (NOAA), at the tide gauge location shown in Fig. 2 (Station ID: 8665530). The still water level time record contained the tidal, sea level rise, and non-tidal residual

*components. The tidal time record was obtained from NOAA's stationary harmonic analysis. The non-tidal residual component was assumed to be equivalent to the storm surge. The storm surge time record was calculated by subtracting the tidal time record from the still water level time record.* *Data for precipitation was obtained at an hourly resolution from the ERA5 reanalysis dataset (Hersbach et al., 2020) at the grid location of 32.75o North, 79.75o West (see Fig. 2). The ERA5 dataset has a spatial resolution of 0.25o, roughly equivalent to 30 kilometers. The time*  *record for the storm surge, tides, and precipitation*  *had an overlapping time record of 24 years and 4 months."*

*L 118 "* *Furthermore, to remove the sea level rise component, the 1-year moving average of the skew surge time record was subtracted from the skew surge time record (Arns et al., 2013). The*  *sea level rise component was also used to identify the current sea level as 0.2 meters above Mean Sea Level (MSL)."*

3.  The uncertainty bands presented in Figure 6 indicate that, in some cases, the return value of a 1000-year event has narrower uncertainty compared to a 10-year event. Does this result only reflect the uncertainty associated with the surrogate damage model for each return period? It is given that a direct comparison of confidence intervals may not be valid among different numbers of stochastic variables, as the stopping criteria are based on uncertainty. Could the authors provide an additional explanation regarding the interpretation of these uncertainty bands? What information do they provide?

    The authors agree that little explanation is provided on the interpretability of the uncertainty bands of Fig. 6.

    The uncertainty bands in Fig. 6 show the 5% and 95% confidence intervals in the return values, which are provided by the TGP-LLM damage and uncertainty response to the d-dimensional space (i.e., the space created by stochastic variables). The purpose of showing the uncertainty bands in Fig. 6 is to give a more interpretable and visual idea of the significant differences between the different risk curves in d-dimensions compared to Figures A5 and A6, which only investigate the mean response of the TGP-LLM. The uncertainty can be used to show that the risk curve in two dimensions significantly differs from all other risk curves when the return period (RP) is larger than 10 years (see L 390). L 402 then gives an explanation to why this is the case.

    Various factors can drive the uncertainty in the damage response. These factors can be systematic or random.

Systematic uncertainty (i.e., bias) is present but is not shown in the uncertainty bands of Fig. 6. The main factors driving this uncertainty are: (1) the digital elevation model, which can cause the damage response to be sensitive to its resolution and accuracy (e.g., Xu et al., 2021) (see L 441); (2) the generation of boundary conditions; which makes the damage response sensitive to how compound flood drivers are parameterized, and subsequently simulated (e.g., Bates et al., 2021); (3) the hydrodynamic model, which can neglect certain physical processes, causing inaccuracies in the flood hazard (see L 434); and (4) the exposure and vulnerability, which can cause sensitivity in the damage response for a constant flood hazard (Jongman et al., 2012). The systematic uncertainty will vary based on the magnitude of the synthetic event. Hawker et al. (2024) achieve these results when comparing two digital elevation models.

The uncertainty bands in Fig. 6 only show factors driving the random uncertainty. The main factors driving this uncertainty are directly related to the ability of the TGP-LLM to fit the damage response to the d-dimensional space. These factors are thus affected by: (1) the size of the d-dimensional space, and (2) the complexity of the damage response surface to the d-dimensional space (see L 372-376). The random uncertainty can be reduced by running additional simulations. However, since the TGP-LLM is regularized, it cannot fit the damage response perfectly, resulting in a portion of this uncertainty always being present (see L 342).

One last factor that can be used to show uncertainty is the statistical uncertainty (e.g., Eilander et al., 2023). Here, it is not quantifiable as only one stochastic event set is included per d-dimensions.

The uncertainty-based stopping criterion ensures the mean Active Learning Mackay (ALM) of non-simulated events is below a value for all dimensions (L 266). Nonetheless, differences in the width of the uncertainty bands can still occur. The authors attribute this to three reasons.

Firstly, a smaller uncertainty can be associated with a synthetic event once simulated. Moreover, the uncertainty bands are associated with the return values, not the events. Thus, on the one hand, events with large return values will show smaller uncertainty. This is because (1) the events have been simulated (either in the initialization (L 260-262) or because there is a high ALM associated with the event), and (2) the uncertainty bands and the damages associated with a RP are likely associated with the same synthetic event as there are large differences in damages between consecutive RPs. On the other hand, events with RPs between 0.5 and 100 years will show larger uncertainties. This is because (1) a large proportion of these events have not been simulated; (2) the logarithmic scale on this x-axis makes it difficult to visualize the uncertainty associated with specific RPs; and (3) the uncertainty associated with a simulated event will be placed at lower RPs, as the differences in damages between RPs will be small.

Secondly, the TGP-LLM can partition the d-dimensional space, which allows the TGP-LLM to model heteroskedasticity (see L 242-245). However, this does not affect Fig. 6, as the TGP-LLM did not have any partitions for all the d-dimensional spaces when the stopping criterion was met.

Thirdly, the predictability of the damage by the TGP-LLM changes depending on the magnitude of the synthetic events and the number of dimensions considered. The less predictable a region of the d-dimensional space becomes, the higher the uncertainty.

The authors made the following changes to the manuscript:

*L 386 "Additionally, since the TGP-LLM is a generative model (i.e., it captures and models the distribution of the output), the  5% and 95% confidence interval associated with  the TGP-LLM damage response to the d-dimensional space is used to show the 5% and 95% confidence interval associated with each RP. "*

*L 390 "The uncertainty bands of the TGP-LLM can be used to show statistically significant differences between the  risk curve  in two dimensions and the risk curves in higher dimensions when the  RP > 10 years ."*

*L 399 "The uncertainty bands in Fig. 6 show the uncertainty driven by the confidence of the TGP-LLM in modeling the damage response to a d-dimensional space.  The uncertainty-based stopping criterion ensures the mean ALM for non-simulated events is at least a certain value for all dimensions. However, differences in the width of the confidence interval at different RPs can be expected because of the following three reasons. Firstly, the uncertainty associated with simulated events will be smaller. This is most noticeable at large RPs, where the density of events is lower in the d-dimensional space, making them more likely to be simulated. Moreover, the uncertainty and damage of a RP are likely to represent the same synthetic event, as the difference in damages between consecutive RPs is large. At smaller RPs, this is less noticeable as: (1) a large proportion of these events have not been simulated; (2) the logarithmic scale on this x-axis makes it difficult to visualize the uncertainty associated with specific RPs; and (3) the uncertainty and damage of a RP may not represent the same synthetic event because of small differences in damages between consecutive RPs. Secondly, the TGP-LLM can partition the d-dimensional space, which allows the TGP-LLM to model heteroskedasticity. However, this did not affect the uncertainty bands in Fig. 6, as the TGP-LLM did not have any partitions for all the d-dimensional spaces when*

*the stopping criterion was met. Thirdly, the predictability of the damage by the TGP-LLM changes depending on the magnitude of the synthetic events and the number of dimensions considered.*

*A stricter stopping criterion will lead to a larger confidence in the risk curve, which could show a larger number of significant differences between the risk curves. However, this will come at the expense of a larger computational cost."*

*L 421 "Active learning allows for the number of numerical simulations to be minimized given an input-to-output relationship. The number of simulations will depend on how complex this relationship is, but as shown in Sect. 3.2, the TGP-LLM is still able to reach the stopping criterion for different outputs that have a different response to the flood drivers. A stricter stopping criterion can be used to achieve a higher confidence in the results. However, this will increase the computational cost. Furthermore, the stopping criterion may never be reached as the TGP-LLM is regularized."*

4. L 207: Authors explain that they used a Gaussian distribution to reconstruct the water levels time series. It would be better to cite a previous study or provide a proper justification for this assumption. Same for the rainfall, how well does their simplification of the hydrograph and hyetographs represent real events? Additionally, how well we can represent real events using a constant rainfall (point estimate) over the entire domain or constant hydrographs along the coast? I think this is an important point that needs to be addressed at least as a limitation of the analysis presented.

The authors agree that Gaussian distributions are not the best approach to modeling the boundary conditions of skew surge and precipitation events, as they are symmetrical, rigid, and monotonically increasing/decreasing before/after the maximum driver magnitude.

An example of a state-of-the-art approach for the time series of the total water level boundary condition would be to use trapezoidal distributions (Poelhekke et al. 2016). These should be modified according to Anderson et al., 2019 to improve the representation of the boundary condition (Marra et al., 2023). These boundary conditions are flexible and unsymmetrical, providing a higher accuracy when comparing the synthetic events with the historical events.

The authors used Gaussian distributions as the parameters are easily interpretable, and the number of parameters required to define one is minimized. Moreover, they can be easily applied to all flood drivers. This enhanced the interpretability of the results. The authors also believe that the generation of stochastic event sets is as important as the generation of the boundary conditions. The approach currently used

by Anderson et al. (2019) uses Gaussian copulas, which are less flexible than the vine copulas used by the authors. Therefore, while the state-of-the-art is more accurate at simulating historical events, the tradeoff for the interpretability and the dependence modeling was too large for the authors.

Spatially homogeneous rainfall conditions underestimate the hazard associated with the pluvial driver if the precipitation magnitude exceeds a threshold (Wang et al. 2022). To counteract this, a point estimate located at 32.75° North, 79.75° West (L 114) was used, which was expected to have historical observations that show the most extreme observations according to NOAA (National Oceanic and Atmospheric Administration (NOAA), n.d.).

Spatially heterogeneous rainfall and total water level boundary conditions are expected to increase the accuracy of the generated events if they can be correctly quantified (e.g., Apel et al., 2016; Bakker et al., 2022). However, the gain in accuracy is expected to be dependent on (1) the size of the hydrodynamic model, and (2) the accuracy and resolution of the data source. Thus, for Charleston County, the gain in accuracy is expected to be marginal. Moreover, spatially heterogeneous boundary conditions would require further parameterization of the boundary conditions, which would increase the problem's dimensionality and by extension, increase the computational time.

Nonetheless, the choice of boundary condition will not affect the use of the conceptual framework as long as the damage response to the *d*-dimensional space is consistent for all synthetic events in the stochastic event set.

Therefore, the authors have added the following text after L 430:

*"The choice of boundary condition will not affect the conclusions drawn from the conceptual framework as long as the damage response to the d-dimensional space is consistent for all synthetic events in the stochastic event set. Nonetheless, the current representation of the boundary conditions for the flood drivers is less accurate compared to the state-of-the-art (e.g., Apel et al., 2016; Bakker et al., 2022; Anderson et al., 2019; Marra et al., 2023). This is because (1) Gaussian distributions force the time series to be symmetrical, rigid, and monotonically increasing/decreasing before/after the peak magnitude of the event, and (2) spatially homogeneous boundary conditions do not represent historical events if the model domain is large and are based on the data from a point source. These boundary conditions were used because they minimize the number of parameters, ensure the results are interpretable, and can be applied to all flood drivers. Moreover, they facilitate the use of vine copulas, which offer more flexibility when inferring the natural variability."*

5. The authors use the SFINCS model, which was set up and validated in a previous study. While this provides a solid foundation, I believe it would be beneficial to include some additional key details about the model setup in this paper. For example, providing information on the model resolution, whether constant or spatially varying roughness was used, and how infiltration was handled (if included).

The authors agree it is important to clarify the data and modeling choices used in the SFINCS model.

The authors propose the following changes:

*L 203 "For more details on SFINCS, see Leijnse et al. (2021) and van Ormondt et al. (2024). For Charleston County, the SFINCS model had a 200x200-meter grid resolution. The native 1x1-meter resolution information for the topo bathymetry and land roughness were included with a subgrid lookup table. The topo bathymetry data was based on the Coastal National Elevation Database (CoNED; Danielson et al., 2016; Cushing et al., 2022). For the spatially varying land roughness, the National Land Cover Database (NCLD; Homer et al., 2020) was used and reclassified to manning roughness values following Nederhoff et al. (2024). Drainage was handled with: (1) pumps located in the Charleston Central sub-county (Diermanse et al., 2023), and (2) the Curve Number infiltration scheme, which was based on the United States General Soil Map (STATSGO2; U.S. Department of Agriculture, 2020) following Nederhoff et al. (2024).*

*. For our application, only two boundary conditions were required: (1) the still water level at the coast, and (2) the precipitation."*

6. L 134: The description of applying a threshold to skew surge magnitude when it "co-occurred with a higher high tide" is confusing. Since the definition of skew surge inherently depends on the high tide (or higher high tide) in a tidal cycle, further clarification would benefit the reader.

In addition to the additional text provided after L 117 to answer comment 1, the authors propose the following changes:

*L 134 "Charleston Harbor's tidal record showed daily inequalities larger than semi-diurnal differences in S.Mags. Therefore, applying Peak Over Threshold (POT) to all S.Mags could have identified high-water events that were not extreme as they could co-occur with lower high T.Mags. Therefore, to only identify extreme high-water events,  was only applied to the S.Mag when it co-occurred with a Higher High (HH) T.Mag. *

7. L 3-5: The phrase "large flood hazard when compared to the sum of the individual drivers" could be explained better.

The authors agree that this particular phrase can confuse the readers.

The spatially varying maximum water depth (or intensity) and frequency define the flood hazard. During a compound flood, the non-linear interactions between flood drivers lead to the generation of what is known as a 'transition zone' (Gori et al. 2020). Because of this zone, the compound flood hazard is larger than the flood hazard created by taking the maximum of each flood hazard generated by its respective flood drivers occurring in isolation from one another.

The authors propose the following changes:

*L 2 "Compound floods result from their co-occurrence and can generate a larger flood hazard when compared to the* synthetic flood hazard generated by the respective flood drivers occurring in isolation from one another. "

8. L 138: Is there a specific rationale for selecting a wider (14-day) de-clustering window? Justifying this choice would strengthen the methodology section.

The authors admit the fourteen-day declustering time window is conservative compared to other studies using Peak Over Threshold (POT) along the United States East Coast, which report a declustering time window of approximately four days (e.g., Martin et al., 2024). For Charleston County, using a declustering time window of four days led to the identification of multiple extreme events embedded in events longer than four days. While the S.Mag was not extreme for more than four days, it remained at large magnitudes below the threshold of 0.32 meters between two identified extreme events. Therefore, a declustering time window of four days identified dependent events, which violated the independent and identically distributed assumption of extreme value theory.

Fourteen days was chosen as it was the largest S.Dur identified before placing an upper bound on all driver durations (see answer to comment 9). This ensures all skew surge events were independent and identically distributed.

The authors recommend the following changes to the manuscript:

*L 137 "A threshold of 0.32 meters relative to MSL and a declustering time window of fourteen days between each extreme S.Mag were chosen. The latter was based on the longest S.Dur before restricting the duration of events (see Sect. 2.2.2). This ensured consecutive extreme events were not embedded in events longer than the declustering time window."*

9.  L 144: The manuscript suggests a minimum duration of 6 days for rainfall and skew surge events. Does it mean that there were no hours with rainfall less than 0.3 mm within 6 days during an event? So all the extreme events were simulated for at least 6 days? Could the authors confirm and clarify?

The authors would like to thank the referee for catching this typo in the manuscript, it should be the "maximum" duration and not "minimum". The authors use the ± three-day window (used to quantify P.Mag at L 142) to place an upper bound on the skew surge and precipitation durations. This is to be consistent with how synthetic events are generated and by extension, limit the maximum possible runtime of SFINCS simulations. This led the skew surge and precipitation events to have a maximal (instead of minimal) duration of six days. For Charleston County, this only affects 9 out of 71 events for skew surge and does not affect any of the 71 precipitation events.

Therefore, the authors suggest the following changes to the text:

*L 141 "All six flood driver parameters had to be quantified for each extreme high-water event. POT was applied on S.Mag in Sect. 2.2.1. For P.Mag, the largest value that co-occurred within ± three days of all identified S.Mag extremes were used. For T.Mag, the co-occurring HH tide with all identified S.Mag extremes were used. The P.Dur and S.Dur were taken as the duration of the P.Mag and S.Mag to continuously remain above a critical value, within the ± three-day window used to quantify P.Mag. a minimal duration of six days. For precipitation and skew surge, the values used to define the duration were 0.3 millimeters per hour and 0.2 meters respectively. The P.Lag was defined as the difference in hours between the S.Mag and P.Mag for each extreme high-water event."*

10. L 352: Please specify whether the given time durations refer to the computational time required "per simulation" or the total time for simulating all events.

The authors agree that the term "computational time" is used loosely after L 346 to describe the computational time of the overall process or that of different components.

Therefore, throughout the text, after L 346, the term "overall computational time" has been used to clarify the manuscript. This is now more in line with the abstract. For example:

*L 77 "Therefore, this study aims to explore active learning to improve the quantification of compound flood risk assessments while limiting the increase in overall computational time."*

*L 346 "Figure 4 shows the computational time associated with both approaches when they sample from the testing event set. The overall computational time can be split into three components: (1) Delft-FIAT, (2) SFINCS, and (3) TGP-LLM. For both*

*approaches, Delft-FIAT is the component that requires the largest computational time. This is caused by the Delft-FIAT model being poorly optimized for Charleston County as it preprocesses the exposure data before each simulation. The added computational time for the active learning approach due to the TGP-LLM is relatively small as the number of simulations is small (2.3 minutes). Therefore, the number of simulations is the main factor influencing the overall computational time. For our experiment, the overall computational time was reduced by a factor of four (95.4 vs. 23.6 minutes)."*

*L 358 "For each, the overall computational time is subdivided into three components: (1) Delft-FIAT, (2) SFINCS, and (3) TGP-LLM. A horizontal red line is provided as a reference, showing the overall computational time of the equidistant sampling approach with two dimensions. Henceforth, this will be referred to as either the equidistant sampling computational time, or the reference computational time."*

11. In Figure 2, ensure that the colors representing coastal and inland counties are correct.

    The authors appreciate the referee for finding this typo. Fig. 2 has now been updated.

**Referee Comment 2**

The paper is well structured, clear and complete in all its parts. Although I am not an expert in active learning, I believe that the different assumptions and the application of the models are scientifically sound.

I believe the methodology is original and I think the paper can be published with some minimal technical corrections.

The authors appreciate the constructive and positive feedback received from the second referee, which will improve the quality of the manuscript.

In the course of the text there is more reference to the output of the procedure. I think I understood that it refers to different geographical areas. If so, I ask the authors to specify explicitly in the text

The authors attempted to show that the methodology can be applied to any general output (geographic location, risk metric, etc.) throughout the text. However, the authors understand this can cause a lack of clarity when interpreting the results.

To improve on this, the authors have improved the explanation of the output in its first two mentions:

[revised manuscript text omitted]

Nederhoff, K., Leijnse, T. W. B., Parker, K., Thomas, J., O'Neill, A., van Ormondt, M., McCall, R., Erikson, L., Barnard, P. L., Foxgrover, A., Klessens, W., Nadal-Caraballo, N. C., and Massey, T. C.: Tropical or extratropical cyclones: what drives the compound flood hazard, impact, and risk for the United States Southeast Atlantic coast?, Natural Hazards, https://doi.org/10.1007/s11069-024-06552-x, 2024.

National Oceanic and Atmospheric Administration (NOAA). n.d. "NOAA Atlas 14 Point Precipitation Frequency Estimates: SC." Precipitation Frequency Data Server (PFDS). Accessed December 16, 2024. https://hdsc.nws.noaa.gov/pfds/pfds_map_cont.html?bkmrk=sc.

Poelhekke, L., Jäger, W. S., van Dongeren, A., Plomaritis, T. A., McCall, R., and Ferreira, : Predicting coastal hazards for sandy coasts with a Bayesian Network, Coastal Engineering, 118, 21–34, https://doi.org/10.1016/j.coastaleng.2016.08.011, 2016.

U.S. Department of Agriculture: U.S. General Soil Map (STATSGO2) for Florida, Georgia, South Carolina, North Carolina and Virginia., https://gdg.sc.egov.usda.gov/, (last access: 8 January 2021), 2020.

van Ormondt, M., Leijnse, T., de Goede, R., Nederhoff, K., and van Dongeren, A.: A subgrid method for the linear inertial equations of a compound flood model, https://doi.org/10.5194/egusphere-2024-1839, 2024.

Wang, H.-J., Merz, R., Yang, S., Tarasova, L., and Basso, S.: Emergence of heavy tails in streamflow distributions: the role of spatial rainfall variability, Advances in Water Resources, 171, 104 359, https://doi.org/10.1016/j.advwatres.2022.104359, 2023.

Williams, J., Horsburgh, K. J., Williams, J. A., and Proctor, R. N. F.: Tide and skew surge independence: New insights for flood risk, Geophysical Research Letters, 43, 6410–6417, https://doi.org/10.1002/2016gl069522, 2016.

Xu, K., Fang, J., Fang, Y., Sun, Q., Wu, C., and Liu, M.: The Importance of Digital Elevation Model Selection in Flood Simulation and a Proposed Method to Reduce DEM Errors: A Case Study in Shanghai, International Journal of Disaster Risk Science, 12, 890–902, https://doi.org/10.1007/s13753-021-00377-z, 2021.